# Real-Time Power Control of Doubly Fed Induction Generator Using Dspace Hardware

**Manale Bouderbala** [1,*] , **Hala Alami Aroussi** [2] , **Badre Bossoufi** [1,*] **and Mohammed Karim** [1]

[1] Faculty of Sciences Dhar El Mahraz, Sidi Mohammed Ben Abdellah University, Fez 30000, Morocco
[2] Ecole Supérieur de Technologie, Mohamed Premier University, Oujda 60000, Morocco
* Correspondence: manale.bouderbala@usmba.ac.ma (M.B.); badre.bossoufi@usmba.ac.ma (B.B.)

**Abstract:** Numerous studies have been undertaken to evaluate wind energy systems' active and reactive power control, the energy produced, and their its link to distribution networks. This research makes a novel contribution to the discipline in this setting. The novelty of this work aims to design a new wind emulator and design a power control approach for a doubly fed induction generator (DFIG)-based wind system. A description of the system was provided first. Secondly, the control strategy was described in detail. Then, it was applied to both converters (machine and grid sides). Three stages were used to evaluate the control solution: (1) a MATLAB/Simulink simulation to validate the reference's persistence (for both real and step wind speeds) and the system's robustness, (2) implementation in real-time on a dSPACE-DS1104 board linked to an experimental laboratory bench, and (3) overlapped comparison experimental and simulated data to conduct a thorough quantitative and qualitative analysis using the root-mean-square error measures. The simulation and experimental findings demonstrate that the suggested model is valid and presents an excellent correlation between experimental and simulated results regarding wind speed variation.

**Keywords:** DFIG; power control; wind emulator; real wind offshore profile

## 1. Introduction

### 1.1. Background

Since 2013, the global offshore industry has grown by 24 per cent per year, bringing total installations to 29.1 GW, accounting for 5% of total worldwide wind capacity by the end of 2019 [1,2]. Offshore wind energy is now the trend in wind energy development since the offshore wind speed is higher than onshore, with a more stable wind direction and low turbulence [3,4].

In 2017, Morocco had a 1015 MW installed wind energy capacity [5]. Its geographical position is strategic: linking the Atlantic Ocean with the Mediterranean Sea qualifies for significant offshore wind potential [3]. Therefore, Morocco provides several suitable zones for the offshore wind energy conversion system (WECS). There is a band of waters at the Atlantic Ocean with wind speeds reaching over nine m/s with a total potential of 135 GW. In addition, there is another band of waters at the northern coast with 43 GW of wind potential [6]. These resources will allow better exploitation of its wind potential and become one of the leading exporters of electricity to Europe, especially Spain and Portugal.

WECS is a turbine that ensures the wind's kinetic energy transformation into mechanical energy. The latter is amplified with the gearbox before its conversion into electrical energy with a generator (As shown in Figure 1).

Wind energy conversion systems rely heavily on control systems. A well-designed WECS control system results in inefficient power generation, high power quality, and reduced aerodynamic and mechanical loads, contributing to the installation's longevity [7]. The wind turbine and the generator must be controlled appropriately for these reasons.

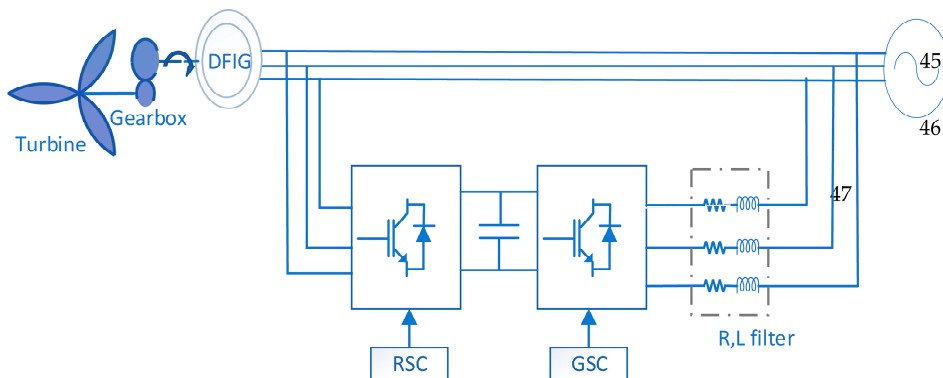

**Figure 1.** Wind energy conversion system.

The turbine control objectives determine the operating time of each control system and each control system and must be clearly established to avoid confusion in the analysis of control methods.

The turbine is controlled by pitch control and MPPT control. The pitch angle is a major parameter of the wind turbine, as it determines the angle of attack of the wind. Thus, the rotation of the blades around their own axis modifies the relative wind flow and, consequently, the aerodynamic loads exerted on the rotor. The "pitch control" command allows the modification of the pitch angle of the turbine blades in order to control their aerodynamic efficiency [8].

Indeed, the first maintains the turbine's distortion at speeds over its nominal value [9], whereas the MPPT extracts the maximum energy [10]. The wind operation zone can be separated into four zones (Figure 2). When the wind speed is inadequate to propel the wind turbine into production, the first Zone occurs. In Zone 2, the electromagnetic torque is adjusted to maximize power generation while keeping a steady blade pitch angle (MPPT algorithm). When the wind velocity increases (Zone 3), the pitch angle should be modified to maintain the nominal power output. The fourth Zone is activated when the wind speed surpasses the nominal power capacity of the turbine. At this point, the emergency mechanisms shut down the turbine to protect the WECS from damage [11].

The authors in [12] present the MPPT algorithm's principle. It is crucial to compute the appropriate electromagnetic torque in real-time, which is utilized to adapt the rotor speed in response to wind speed by tracking the optimal turbine speed that results in a power coefficient Cp = 0.5 [13].

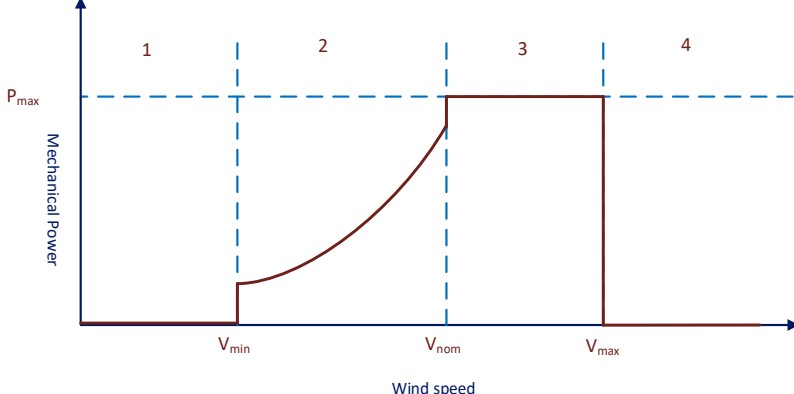

**Figure 2.** WT characteristics [14].

Ugalde-Loo et al. provide an overview of the various generators employed in the WECS system in machine control [15]. The generator used in this work is a doubly fed induction generator (DFIG), which improves system efficiency by ensuring a proper ro-

tational speed, decreasing noise and mechanical stress, improving power quality, and adjusting for torque and power pulsations [16,17].

Wind power is based on a random resource that can negatively impact the grid's stability [18]. Therefore, it is essential to regularly control the energy transfer to the grid for minimal losses. Since an AC-DC-AC converter controls the system, the rotor side control guarantees an excellent tracking of powers. At the same time, the grid side control ensures voltage and frequency regulation and DC bus stability. This approach needs to consider all the machine's parameters instead of a simplified DFIG model [7].

Validating and enhancing the control of wind turbines requires a test bench environment. Wind turbines are known to exhibit nonlinear behavior. A wind turbine emulator is an essential tool for modeling a real wind turbine's static, dynamic, and nonlinear properties without relying on available natural wind resources or commercial wind turbines. The authors in [19,20] used the Dspace card to realize a hardware under-loop simulation. As a perspective in their study, authors in [21] plan to conduct experimental tests after realizing the hardware in the loop.

### 1.2. Motivation and Contribution

This article conducts an experimental investigation of a vector control technique based on a proportional-integral controller. First, the control approach is integrated into an experimental test bench through a DSpace DS1104 board to maintain the voltage and frequency of the DFIG stator outputs within their acceptable operating parameters. The authors offer the notion of a new wind emulator and a test bench for evaluating control tactics, which they demonstrate using a real-time offshore wind profile. Finally, a simulation is created in the Matlab/Simulink environment using the real-time interface (RTI) to compare the performance and resilience of the proposed control model to experimental results.

The novelty of this work lies in the methodology relative to the comparison between simulation and practical results. In fact, the last studies [22–25] used only a qualitative analysis to validate their results. Their concern was about having the same shape for simulation as for experimental results. The qualitative analysis is considered unstructured. It is known for being subjective, individualized, and personalized. Because of this, qualitative data are inferior if they are the only data in the study. On the other hand, quantitative analysis is structured and accountable. This type of data is formatted so it can be organized, arranged, and searchable. Because quantitative data are more concrete, we have considered using RMSE (root mean square error) to analyze better the margin of errors between simulation and experimental results.

This essay is organized as follows. Section 2 describes the system's components, including wind turbine, DFIG, and back-to-back converters. Then, in Section 3, the DFIG control approach is demonstrated, created, and designed. Section 4 describes in detail the testbench for a new wind emulator. The experimental data are given, debated, and utilized to validate the performance of the simulation results. Finally, Section 6 summarizes the article by offering some concluding thoughts and suggestions for further research.

## 2. Description of the System

### 2.1. Wind Turbine

Electricity is generated using an alternator driven by a turbine. Y. Charabi et al. [26] describe transforming the wind's kinetic energy into electrical energy using various wind turbines. The wind's kinetic energy generates wind energy. The mathematical modelling of the turbine allows for the aerodynamic power ($P_{aer}$) is given in (1) [27].

$$P_{aer} = C_P(\lambda, \beta).\frac{\rho.S.V^3}{2} \tag{1}$$

According to Betz's law, no wind turbine can extract 100% of its energy; thus, the power coefficient $C_p$ is a function of $\lambda$, representing the turbine's tip speed ratio and $\beta$, the

blades' orientation angle. Therefore, the following generic Equation is utilized in (2) [28,29] to represent the power coefficient.

$$C_p(\lambda, \beta) = (0.5 - 0.0167.(\beta - 2)).\sin\left(\frac{\pi.(\lambda + 0.1)}{18 - 0.3(\beta - 2)}\right) - 0.00184.(\lambda - 3).(\beta - 2) \quad (2)$$

The dynamic equation of the wind turbine is given in (3):

$$J\frac{d\omega_{mec}}{dt} = T_{mec} = T_g - T_{em} - f.\omega_{mec} \quad (3)$$

The gearbox gain G used to adjust the turbine's speed to the DFIG is given in (4)

$$G = \frac{T_t}{T_g} = \frac{\omega_{mec}}{\omega_t} \quad (4)$$

### 2.2. Doubly Fed Induction Generator

The doubly fed induction machine (DFIM) is a wound rotor induction machine that resembles a squirrel cage machine in structure. It comprises two sets of three-phase windings: one for the stator and another for the rotor [30–32].

The DFIG mathematical model expresses the stator and rotor phase voltages and flows as functions of the currents that traverse them [33].

Where $\omega_s$ is defined as synchronous speed, $\omega_{mec}$ is the mechanical speed with

$$\omega_r = \omega_s - \omega_{mec} \quad (5)$$

$V_{s(d,q)}$, $V_{r(d,q)}$ are the stator and rotor voltages, respectively, in the dq Park frame. $\Phi_{s(d,q)}$, $\varphi_{r(d,q)}$ are the stator and rotor fluxes, respectively, in the dq Park frame. $I_{s(d,q)}$, $I_{r(d,q)}$ are the stator and rotor currents, respectively, in the dq Park frame.

$$\begin{cases} V_{s(d,q)} = R_s.I_{s(d,q)} + \dfrac{d\varphi_{s(d,q)}}{dt} \mp \varphi_{s(q,d)}.\omega_s & (6) \\ V_{r(d,q)} = R_r.I_{r(d,q)} + \dfrac{d\varphi_{r(d,q)}}{dt} \mp \varphi_{r(q,d)}.\omega_r & (7) \end{cases}$$

$$\begin{cases} \varphi_{s(d,q)} = L_s.I_{s(d,q)} + L_M.I_{r(d,q)} & (8) \\ \varphi_{r(d,q)} = L_r.I_{r(d,q)} + L_M.I_{s(d,q)} & (9) \end{cases}$$

Equation (10) describes the electromagnetic torque:

$$T_{em} = p\left(\varphi_{sd}.I_{sq} - \varphi_{sq}.I_{sd}\right) \quad (10)$$

The machine is composed of two parts: stator and rotor resistances (Rs and Rr, respectively), the stator, the rotor, and the mutual inductances ($L_s$, $L_r$ and $L_M$, respectively).

The stator and rotor powers are expressed as follows:

$$\begin{cases} "P_s = V_{sd}.I_{sd} + V_{sq}.I_{sq}" & (11) \\ "Q_s = V_{sq}.I_{sd} - V_{sd}.I_{sq}" & (12) \end{cases}$$

$$\begin{cases} "P_r = V_{rd}.I_{rd} + V_{rq}.I_{rq}" & (13) \\ "Q_r = V_{rq}.I_{rd} - V_{rd}.I_{rq}" & (14) \end{cases}$$

The corresponding circuit of the DFIG in the reference frame is seen in Figure 3. (d-q).

### 2.3. Back-to-Back Converter

Without the electronic converter, the rotor cannot be connected to the grid. As a result, it manages the quantity of electricity delivered to the grid and adapts the frequency to the grid's requirements. Its converter generates around 30% of the DFIG's total power. The

DFIG's rotor is frequently connected to the grid in a back-to-back configuration, enabling four-quadrant operation. In this design, a DC bus connects two voltage source converters. The diagram illustrates the structure utilizing a two-level PWM converter. The control signals are supplied by a sinusoidal PWM (SPWM) (voltage references compared to a triangular carrier at switching frequencies) [12].

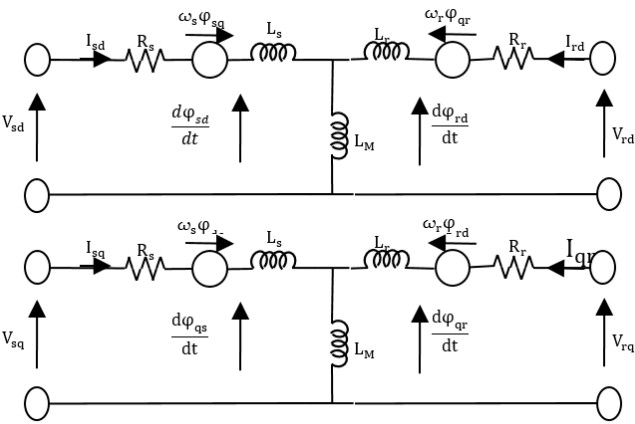

**Figure 3.** DFIG modelling.

The control is applied to the AC-DC-AC converter, as seen in Figure 4. The back-to-back comprises an inverter, a rectifier, and a direct current link. Each leg of the IGBT transistor can be interpreted as a dual-way switch: the two IGBT cannot have the same state (ON/OFF) simultaneously to avoid any potential electrical short circuit.

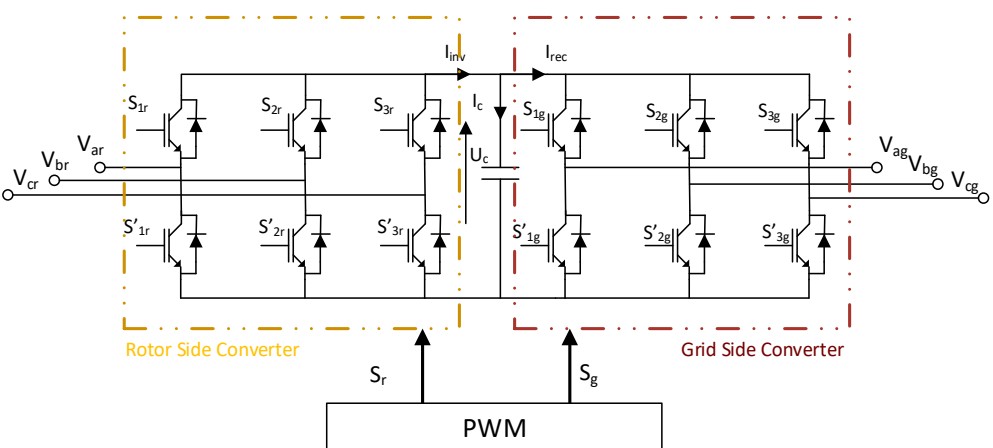

**Figure 4.** Back-to-back converter.

The following relation express the current in the capacitor:

$$I_c = I_{inv} - I_{rec} \tag{15}$$

## 3. DFIG Control

Vector control is used to manage the generator's active and reactive powers independently and to ensure their decoupling due to the stator's flux orientation. When two phases d-q are provided, the stator flux is oriented parallel to the d-axis, such that the d-axis is parallel to the stator flux vector's direction (Figure 5). The flow expression is transformed into:

$$\varphi_{sd} = \varphi_s, \ \varphi_{sq} = 0 \tag{16}$$

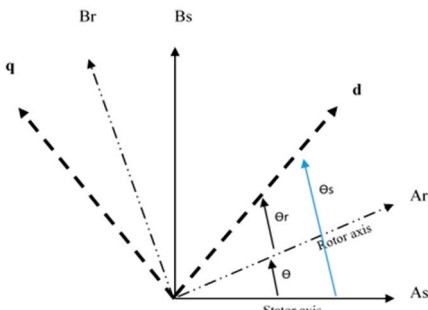

**Figure 5.** FOC orientation.

By injecting Equation (16) into Equation (6) and ignoring the resistance $R_s$, the stator voltages become as follows:

$$V_{sd} = 0, \ V_{sq} = V_s \tag{17}$$

$\theta_s$: Dephasing between the stator and the rotating reference dq.
$\theta_r$: Dephasing between the rotor and the rotating reference dq.
$\theta$: Dephasing between the stator and the rotor.

The phase-locked loop (PLL) was utilized to determine the angles ($\theta_s$, $\theta_r$) required for the transformation of the stator variables and rotor variables. This PLL enables precise estimation of the grid's frequency and amplitude [34].

As illustrated in Figure 6, the controller's architecture is based on a three-phase model of the wind energy system's electromechanical conversion chain [35]. Three commands are required: To begin, MPPT control is used to maximize wind energy extraction, followed by RSC control of the DFIG stator active and reactive powers. Finally, GSC control is accomplished through voltage regulation of the DC bus and the interchange of active and reactive power with the grid [36].

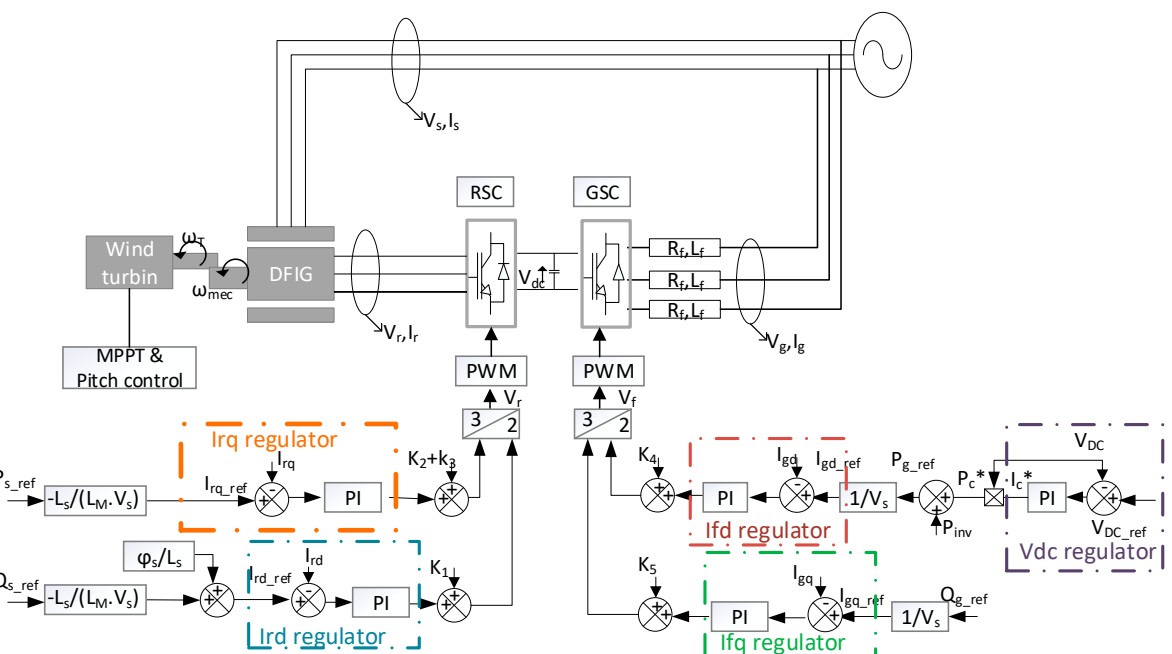

**Figure 6.** System control.

*3.1. Rotor Side Control*

The rotor side converter's control principle is presented in Figure 7.

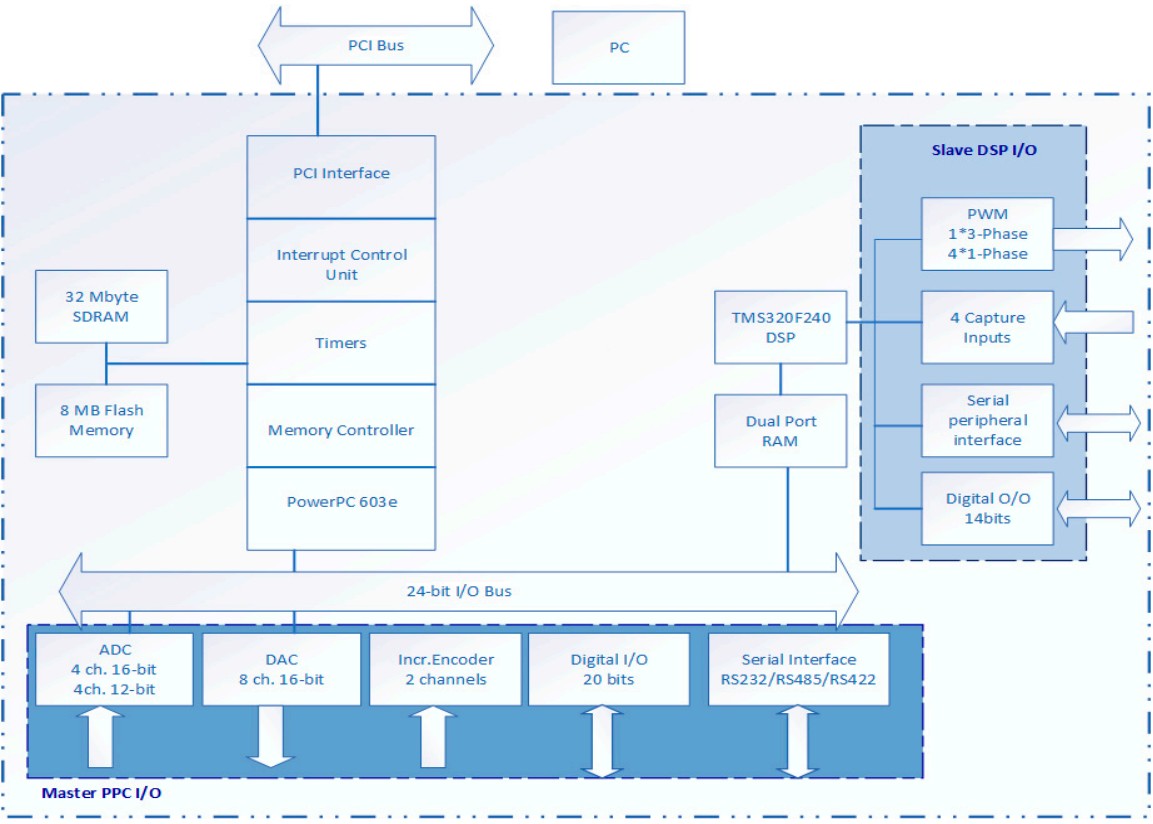

**Figure 7.** Dspace1104 Block Diagram [32].

By injecting (16) into (8), the stator currents are expressed in (18) as follows:

$$\begin{cases} I_{sd} = \dfrac{\varphi_s - L_M.I_{rd}}{L_s} \\ I_{sq} = \dfrac{-M.I_{rq}}{L_s} \end{cases} \tag{18}$$

By replacing (18) with (9), the rotor fluxes are shown in (19) as follows:

$$\begin{cases} \varphi_{rd} = \sigma.L_r.I_{rd} + \dfrac{L_M}{L_s}.\varphi_s \\ \varphi_{rq} = \sigma.L_r.I_{rq} \end{cases} \tag{19}$$

with $\sigma = 1 - \dfrac{L_M^2}{L_s.L_r}$

By injecting the rotor fluxes obtained above in (19) into (7), rotor voltages in d-q frame are expressed in (20) as follows:

$$\begin{cases} V_{rd} = R_r.I_{rd} + \sigma.L_r.\dfrac{dI_{rd}}{dt} - \sigma.L_r.I_{rq}.\omega_r & (20) \\ V_{rq} = R_r.I_{rq} + \sigma.L_r.\dfrac{dI_{rq}}{dt} + \sigma.L_r.I_{rd}.\omega_r + \dfrac{L_M}{L_s}.\varphi_s.\omega_r & (21) \end{cases}$$

The active and reactive powers of the stator as indicated in (22) and (23) are obtained by injecting (17) and (18), respectively, into (11) and (12):

$$\begin{cases} P_s = -V_{sq}.\dfrac{L_M}{L_s}.I_{rq} & (22) \\ Q_s = V_{sq}.\dfrac{\varphi_s}{L_s} - V_{sq}.I_{rd}.\dfrac{L_M}{L_s} & (23) \end{cases}$$

It is worth noting that the powers are mutually exclusive. While the active power is proportional to the quadrature rotor current, the reactive power is not. Controlling the stator's active and reactive power may now be accomplished by regulating the rotor's (dq) axis currents.

However, to express reference currents in d-q frame, one must set $P_s$ and $Q_s$ in (22) and (23) to reference. The resulting $I_{rq\_ref}$ and $I_{rd\_ref}$ are expressed in (24) and (25), as follows:

$$\begin{cases} I_{rq\_ref} = -\dfrac{L_s}{L_M.V_{sq}}.P_{s\_ref} & (24) \\[3mm] I_{rd\_ref} = \dfrac{\varphi_s}{L_s} - \dfrac{L_s}{L_M.V_{sq}}.Q_{s\_ref} & (25) \end{cases}$$

$I_{rd}$ and $I_{rq}$ are set in their references to express rotor voltages in their references. From (20) and (21), $V_{rd\_ref}$ and $V_{rq\_ref}$ are defined in (26) and (27), as follows:

$$\begin{cases} V_{rd\_ref} = (R_r + S.\sigma.L_r).I_{rd\_ref} + k_1 & (26) \\[2mm] V_{rq\_ref} = (R_r + S.\sigma.L_r).I_{rq\_ref} + k_2 + k_3 & (27) \end{cases}$$

with: $k_1 = -\omega_r.\sigma.L_r.I_{rq}$, $k_2 = \omega_r.\sigma.L_r.I_{rd}$, $k_3 = \omega_r.\frac{L_M}{L_s}.\varphi_s$.

As can be seen, the rotor current in the d-q frame is negative. It can have its regulator with a transfer function expressed in (28) (orange and blue rectangle shown in Figure 7):

$$H(s) = \frac{1}{R_r + S.\sigma.L_r} \tag{28}$$

Figure 6 describes the RSC control block diagram. The decoupling and compensating terms in this block diagram are used to manage the active and reactive stator powers by managing the currents in the d-q frame.

*3.2. Grid Side Control*

Once again, Figure 6 illustrates the control principle for the grid side converter, which performs two distinct functions [37]: To begin, control the current flowing through the RL filter; second, control the voltage on the DC bus. The q-axis is considered to be aligned with the grid voltage's angular position. The grid voltages become:

$$V_{gd} = 0 \text{ and } V_{gq} = V_s \tag{29}$$

The voltage at the converter output in the d-q frame is expressed in (30) and (31) as follows:

$$\begin{cases} V_{fd} = V_{gd} + R_f.I_{fd} + L_f.\dfrac{dI_{fd}}{dt} - I_{fq}.L_f.\omega_s & (30) \\[3mm] V_{fq} = V_{gq} + R_f.I_{fq} + L_f.\dfrac{dI_{fq}}{dt} + I_{fd}.L_f.\omega_s & (31) \end{cases}$$

By taking into consideration the assumption of (29), the converter voltage can be expressed as follows:

$$\begin{cases} V_{fd} = (R_f + s.L_f).I_{fd} + k_4 & (32) \\[2mm] V_{fq} = (R_f + s.L_f).I_{fq} + k_5 & (33) \end{cases}$$

with: $k_4 = -I_{fq}.L_f.\omega_s$ and $k_5 = V_{sq} + I_{fd}.L_f.\omega_s$.

It can be seen that the converter current for each axis of the d-q frame can have its own regulator with a transfer function expressed in (34) (red and green rectangle shown on Figure 7):

$$H(s) = \frac{1}{R_f + S.L_f} \tag{34}$$

The active and reactive grid powers are expressed in (35) and (36)

$$\begin{cases} P_g = V_{gd}.I_{gd} + V_{gq}.I_{gq} & (35) \\ Q_g = V_{gq}.I_{gd} - V_{gd}.I_{gq} & (36) \end{cases}$$

The active and reactive grid powers given in (37) and (38) are obtained by injecting (29) into (35) and (36):

$$\begin{cases} P_g = V_s\,.I_{gq} & (37) \\ Q_g = -V_s.I_{gd} & (38) \end{cases}$$

It can be seen $I_{gd}$ and $I_{gq}$ are proportional to the powers $P_g$ and $Q_g$, respectively.

Active and reactive power references $P_{g\_ref}$ and $Q_{g\_ref}$ can be applied, as well as the following reference currents given in (39) and (40), for each axis of d-q frame:

$$\begin{cases} I_{gq\_ref} = \dfrac{P_{g\_ref}}{V_s} & (39) \\[3mm] I_{gd\_ref} = \dfrac{Q_{g\_ref}}{V_s} & (40) \end{cases}$$

The current $I_{gd\_ref}$ is obtained directly from $Q_{g\_ref}$; on the other hand, the current $I_{gq\_ref}$ is achieved by adjusting the DC link voltage.

The powers linking the Dc bus to the rectifier and the inverter are given in (41)–(44) [31,38].

$$P_{inv} = V_{DC}.I_{inv} \tag{41}$$

$$P_{rec} = V_{DC}.I_{rec} \tag{42}$$

$$P_c = V_{DC}.I_c \tag{43}$$

$$P_{rec} = P_{inv} \tag{44}$$

$P_g$ and $P_{rec}$ are set equal by neglecting losses in the converter, RL filter and the capacitor

$$P_{rec} = P_g = V_{DC}.I_{rec} = V_{sq}.I_{fq} \tag{45}$$

The regulation of the power $P_g$ makes the capacitor power $P_c$ controllable. Consequently, the DC bus voltage can be controlled. To determine $P_{g\_ref}$, the powers $P_{inv}$ and $P_c$ need to be known. Equation (46) shows that the capacitor's reference power is proportional to the reference current:

$$P_{c\_ref} = V_{DC}.I_{c\_ref} \tag{46}$$

Figure 6 illustrates how we can manage the DC bus voltage via an external loop (purple rectangle), utilizing a PI controller to generate the reference $P_{c\_ref}$. Additionally, the graphic depicts the GSC control block diagram. This block diagram includes decoupling and compensating terms that allow for independent control of the currents in the d-q frame flowing through the RL filter and the active and reactive powers exchanged between the grid and the GSC.

## 4. Experimental Test Bench

The Dspace card used is the Ds1104 R&D; it is a standard card that can be plugged into a PCI slot of the PC. The Ds1104 is designed for many sectors (engineering, automation . . . ) that demand advanced multivariable control and real-time simulations. It is the product of a combination of an MPC8240 main processor, PowerPC 603e core, 250 MHz internal clock and 8 MB Flash and 32 MB SDRAM memory, with a slave DSP based on a Texas Instruments microcontroller type TMS320F240 [24]. It is a board that converts Simulink blocks to machine code for execution on a DSP-based system.

The prototyping then goes through three steps:

1.  Construction of the control system using Simulink blocks.
2.  Simulation of the system to see the results in different scenarios.
3.  Real-time execution of the model through the DS1104 board.

As shown in Figure 7, the Dspace 1104 R&D controller Board includes ADCs, DACs, digital inputs and outputs and incremental encoders. The DS1104 board is also equipped with a slave DSP (the TMS320F240 DSP) to generate the PWM signals to control the DFIG rotor. Finally, an isolation and adaptation board is also needed.

The hardware component includes a DSpace 1104R&D controller board, a Danfoss VSD, an IGBT inverter, a 1.5 Kw doubly fed induction generator, and a 1.5 Kw Squirrel-Cage-Induction-Machine. The inverter is used to operate the DFIG rotor using the control algorithm. In comparison, the control portion of the system is built using Matlab/Simulink and the Real-Time Interface (RTI) via the Control Desk interface.

The real-time control law implementation steps are as follows:

*   Realize and test the program in the Matlab Simulink environment.
*   Execute the program and generate the C code.
*   Create the environment and the real-time experiment's graphical interface (Layout).
*   Create the link between Simulink and Layout.

The wind emulator was developed by applying the necessary wind profile via a Danfoss (variable speed drive) and a Dspace1104 to a Danfoss (variable speed drive). The SCIG is then told, in concert with the DFIG, to obtain a mechanical speed that matches to the wind profile employed. Figure 8 illustrates the test bench in detail.

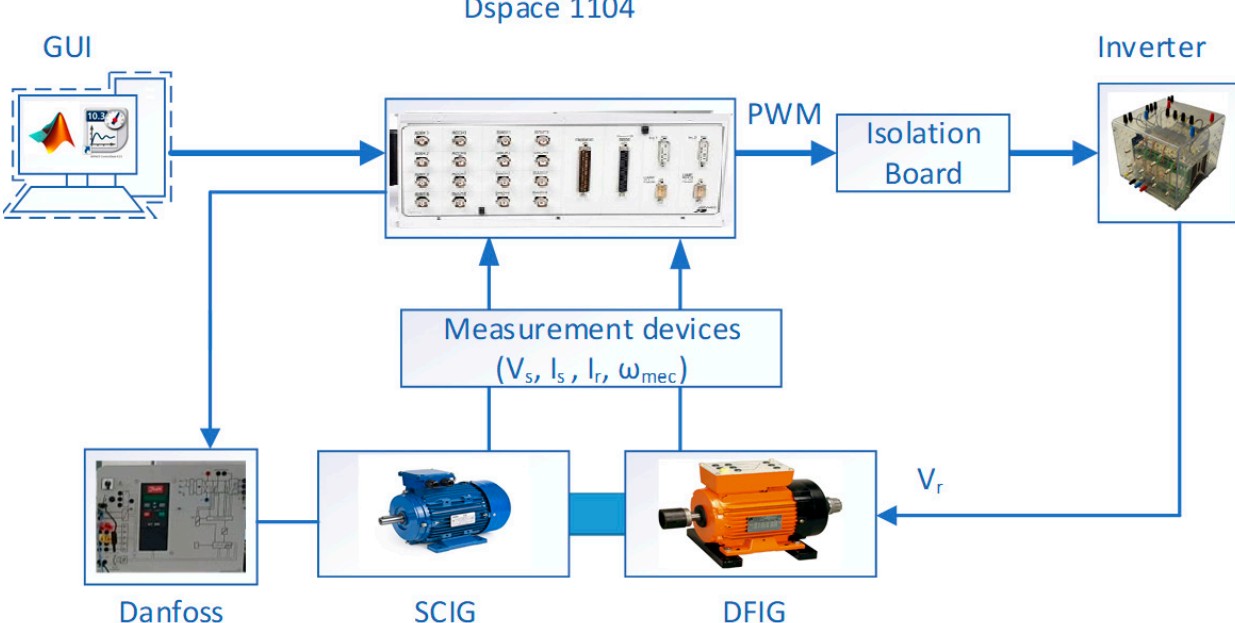

**Figure 8.** Test bench synopsis.

System variables are monitored by using Control Desk software. The graphical interface designed in Control Desk allows to:

Execute the program provided to the DSP with the Layout as GUI.

Test the connections between the Layout and the Simulink variables.

Control, visualize and modify in real-time the different variables of the program developed in Simulink.

To validate and test the created control, an experimental test bench was constructed in the LGEM laboratory, as illustrated in Figure 9.

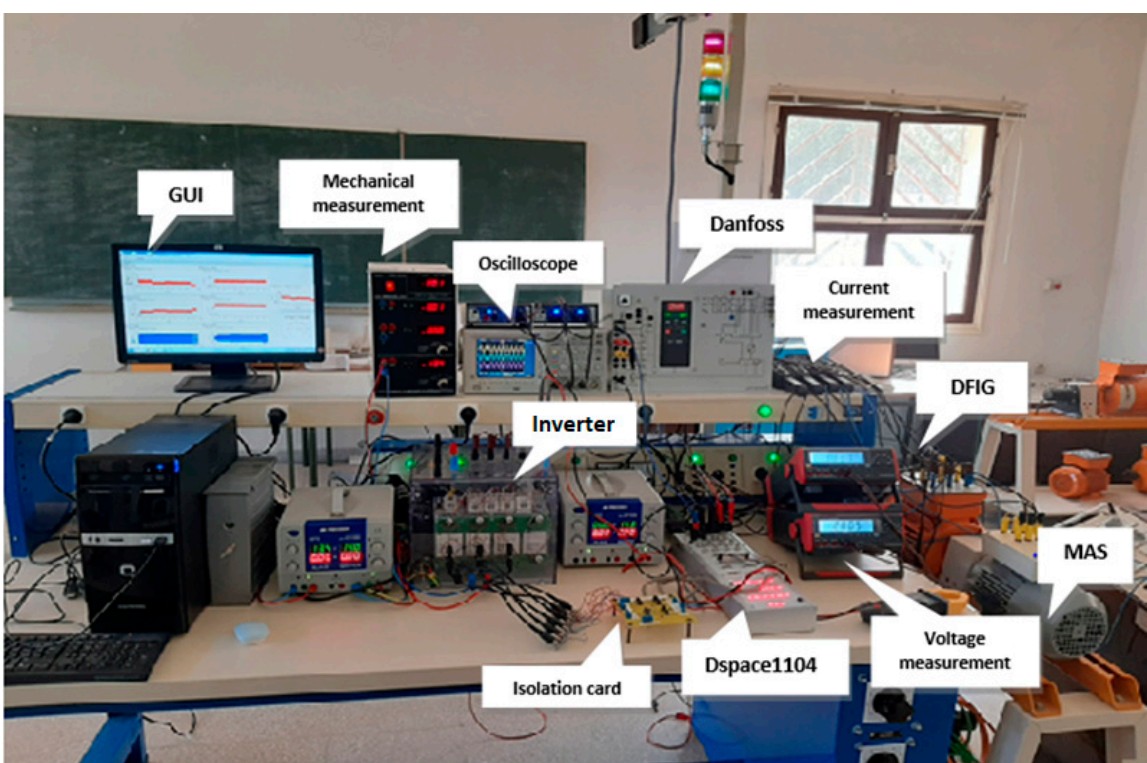

**Figure 9.** The experimental test bench.

## 5. Results and Discussion

Simulation tests were conducted on the wind power system using the MATLAB /SIMULINK tool, which assessed how the control affects system performance and evaluates the control's efficiency, robustness, and output quality. Experimentation on a test bench proved the effectiveness of the vector control method.

### 5.1. Simulation Results

To begin, a simulation of the control method's impact on the system was performed using a step as the wind profile (Figure 10) to demonstrate the robustness and the control's response to abrupt changes. Then, the real offshore northern region of Morocco wind's profile [39,40] (Figure 11) was applied to evaluate the control's efficiency and robustness. Finally, the step size was set to T = $1 \times 10^{-4}$ s (Table A1).

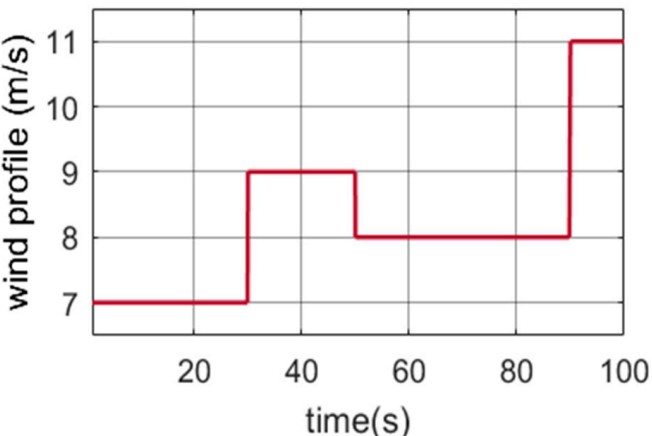

**Figure 10.** Wind step profile.

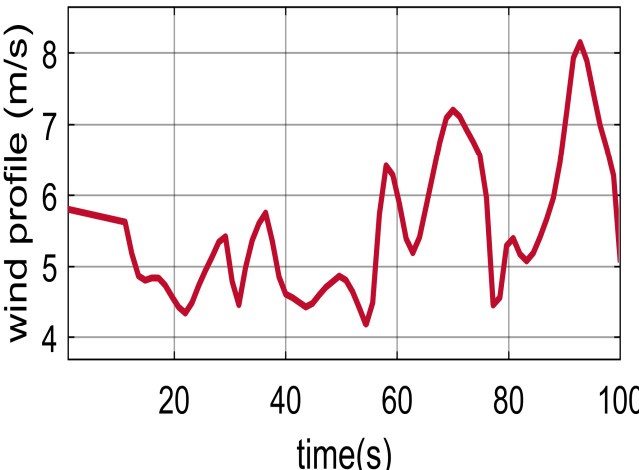

**Figure 11.** Real wind profile.

Figures 12a and 13a depict the active and reactive powers for a step wind profile and a real wind profile. Reactive power was set to zero for a unit power factor. Both active and reactive abilities were consistent with their references. Because the equipment was in generator mode, active power was inversely proportional to the wind shape (Figures 10 and 11). When a random portion of the resulting active and reactive powers is magnified, it becomes clear that the chase is always guaranteed without overshoot, regardless of whether the wind changes are gradual or rapid. Figures 12b and 13b illustrate the quadrature rotor current. It is the inverse of the active power curve (Figures 12a and 13a), because Ps changes linearly with Irq via the negative sign defined in Equation (24); the direct rotor current is illustrated in Figures 12c and 13c. Because Qref is zero, it has a constant value of approximately 5 A, and the remaining constant ratio $\varphi_s$/Lm Equation also equals 5 A.

Since the current is the power image, Figures 12d and 13d display stator currents with an inverse form identical to the wind profile. While the current profile varies, it retains a sinusoidal shape with a period of 0.02 s (independently of the wind shape) corresponding to the grid frequency of 50 Hz. Therefore, the normalized THD value must be less than 5% to have high-quality electricity. In this study, the THD was less than three per cent (Figures 12g and 13g) for both step and real wind profiles, indicating an outstanding power quality. In terms of rotor currents, Figures 12e and 13e demonstrate that the profile fluctuates sinusoidally and proportionally to the electromagnetic torque.

To evaluate the suggested control scheme's performance, which is primarily concerned with tracking active and reactive power references via the management of quadrature and direct rotor currents, (47) uses the root-mean-square error (RMSE) as a performance metric:

$$\text{RMSE} = \sqrt{\sum_{k=1}^{k} y(k) - y_{ref}(k)^2 / k} \tag{47}$$

To test the steady-state tracking performance of the command, the errors of each key parameter were calculated using (47) for both the step and the real wind profile. For the step wind profile, Ps, Qs, Iqr and Idr are tracking their references with an error of 7.7%, 10.5%, 9.7%, and 8.1%, respectively. As for the real wind profile, the corresponding tracking errors are 4.2%, 9.9%, 7.5%, and 4.7%, respectively. The steady-state tracking errors of all parameters are inferior to the real wind profile ones. This is because the pursuit of the reference is better for smooth wind changes than for sudden changes (Figure 14).

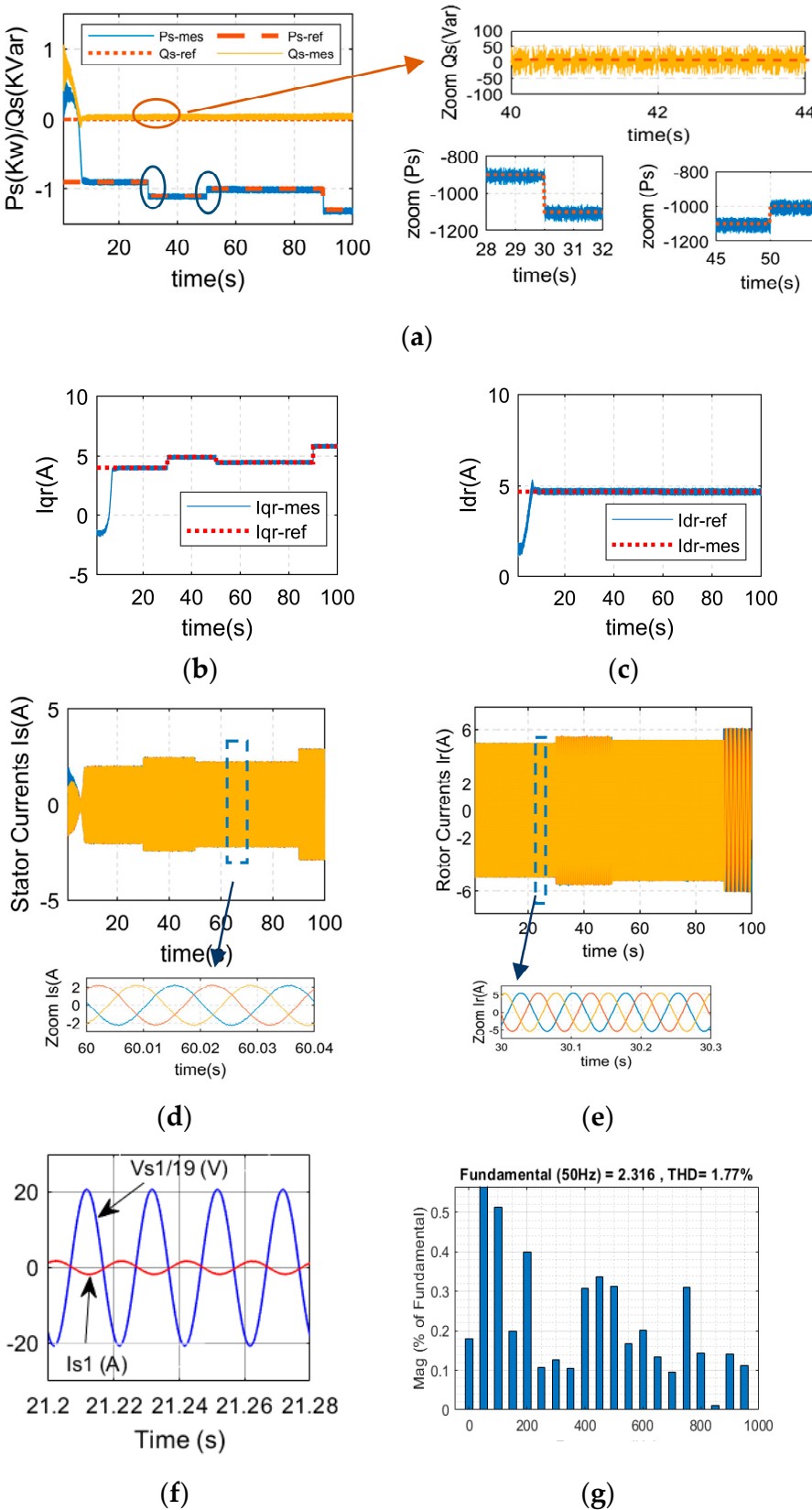

**Figure 12.** Simulation results with step reference. (**a**) Active and reactive powers; (**b**) quadrature rotor current; (**c**) direct rotor current; (**d**) stator currents; (**e**) rotor currents; (**f**) Voltage and current phase a; (**g**) the current.

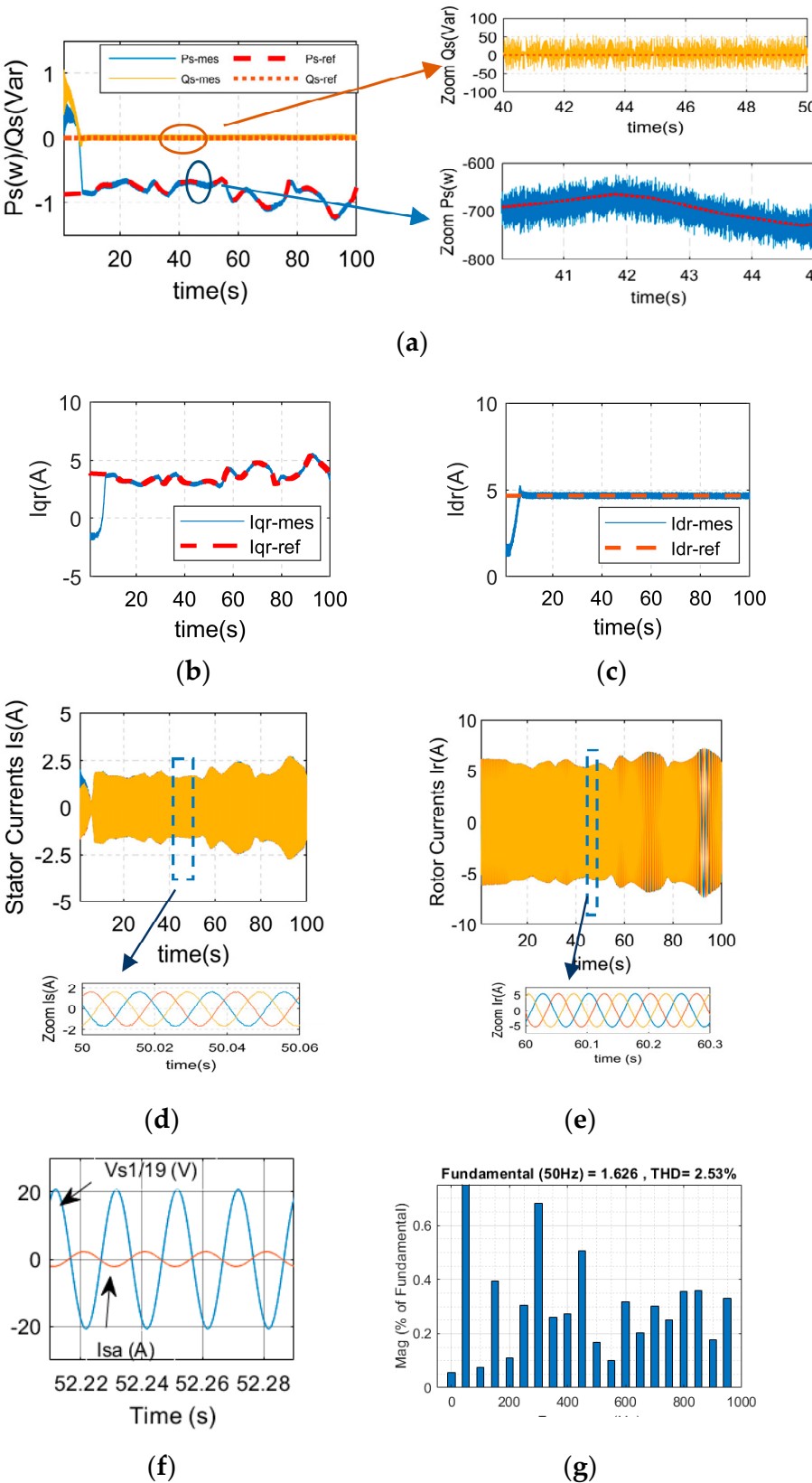

**Figure 13.** Simulation results with real wind profile reference. (**a**) Active and reactive powers; (**b**) quadrature rotor current; (**c**) direct rotor current; (**d**) stator currents; (**e**) rotor currents; (**f**) voltage and current phase a; (**g**) the current.

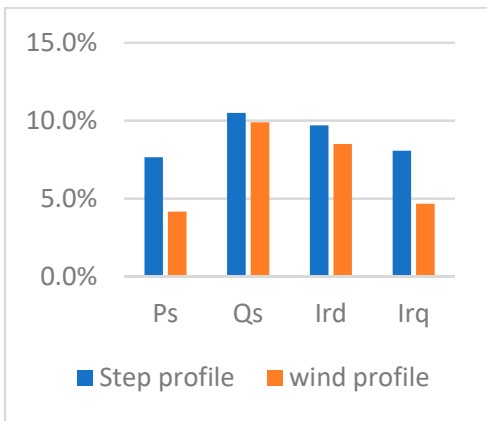

**Figure 14.** RMSE (Simulation).

### 5.2. Experimental Validation

To demonstrate the vector control approach's efficiency on the test bench, identical condition tests were used. System robustness is shown in the first test, while tracking efficiency and control in the face of a changeable wind profile are demonstrated in the second test. The experimental results for the two tests exported from the DS1104 R&D Controller Board via the RTI Control Desk are depicted in Figures 15 and 16.

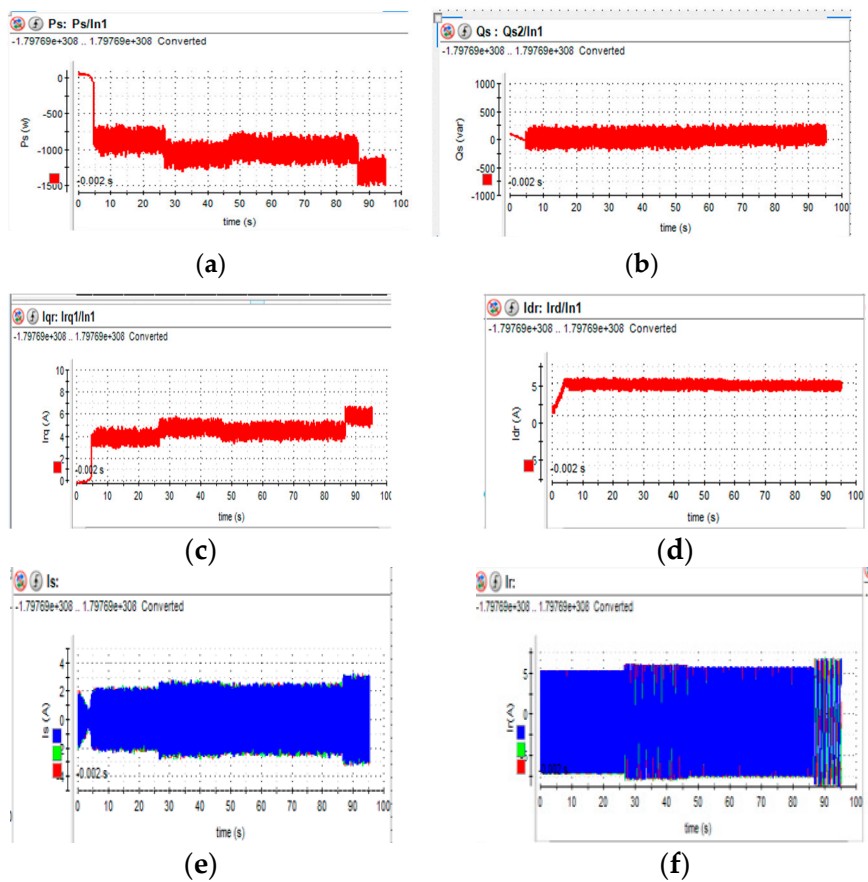

**Figure 15.** Experimental results for a step profile on ControlDesk layout. (**a**) +Active+power+; (**b**) +Reactive+power+; (**c**) +Quadrature+rotor+current+; (**d**) +Direct+rotor+current+; (**e**) +stator+currents+; (**f**) +rotor+currents+.

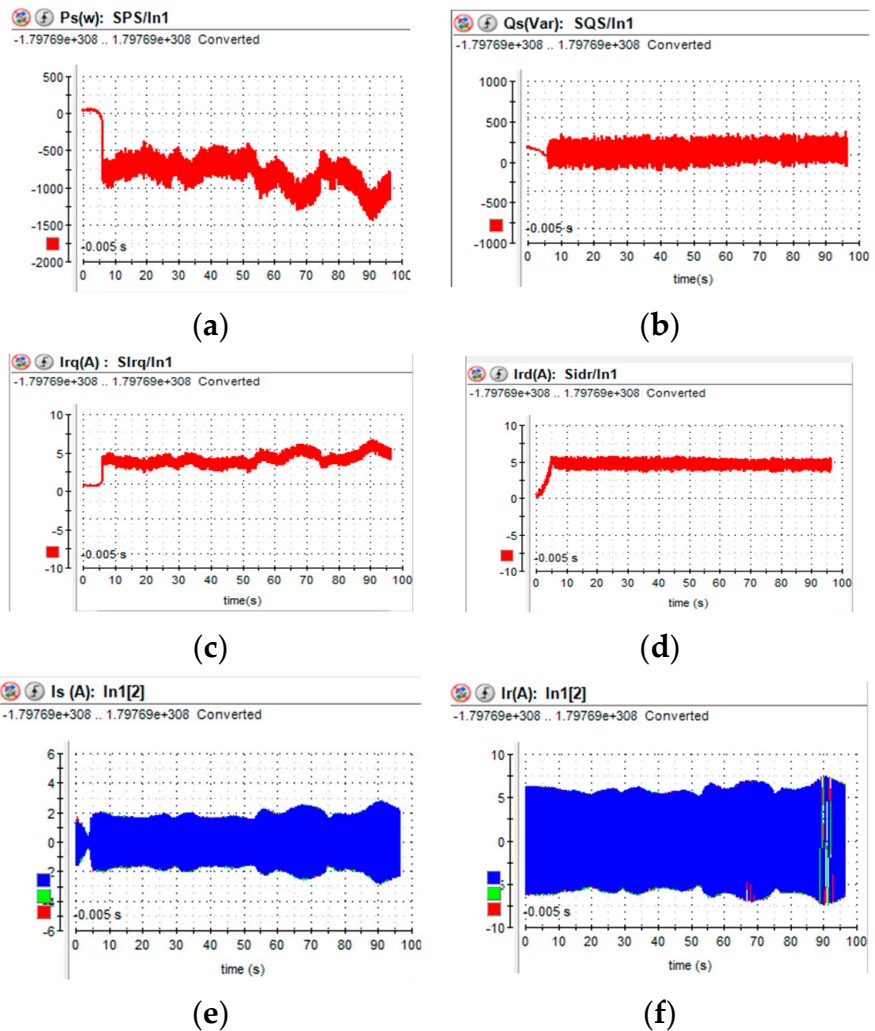

**Figure 16.** Experimental results for a step profile on ControlDesk layout. (**a**) +Active+power+; (**b**) +Reactive+power+; (**c**) +Quadrature+rotor+current+; (**d**) +Direct+rotor+current+; (**e**) +stator+currents+; (**f**) +rotor+currents+.

The experimental demonstration of the control approach's effect on the system uses a step as the step wind profile (Figure 10) to demonstrate the control's response to discontinuities. Then, the control's efficiency and resilience were evaluated using Morocco's wind profile's real offshore northern region (Figure 11).

Because the machine is in generator mode, active power is inversely proportional to the wind shape (Figures 15a and 16a). As can be shown, the pursuit is always assured without overshooting for both variable and steady-state wind fluctuations. The quadrature rotor current is seen in Figures 15c and 16c. Ps has the inverse shape of the active power curve since it varies linearly with Irq via the negative coefficient specified in (24). (Figures 15a and 16a). Figures 15d and 16d illustrate the direct rotor current. Since Qref is zero, it has a constant value of approximately 5 A and the remaining constant ratio $\varphi_s/Lm$ in Equation equals 5 A.

Stator currents are sinusoidal with a constant period of 0.02 s and a frequency of 50 Hz, as seen in Figures 15e, 16e and 17. Additionally, sinusoidal rotor currents are depicted in Figures 15f, 16f and 18. The voltage and current of a single phase are shown in Figure 19. They are in opposition, sinusoidal, and have a frequency of 50 Hz, which is expressed in time units of 0.02 s, ensuring a unit power factor. The MLI control signals applied to the converter are depicted in Figure 20.

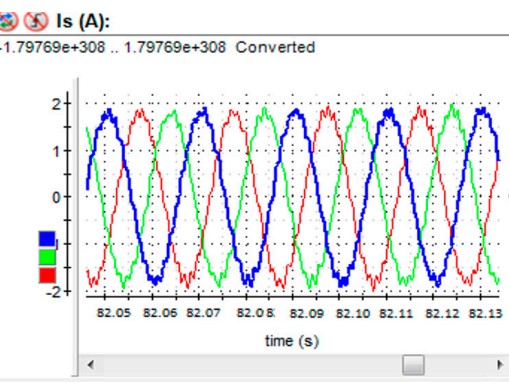

**Figure 17.** Zoom of Is.

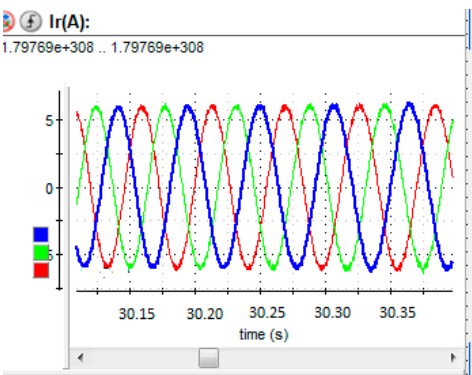

**Figure 18.** Zoom of Ir.

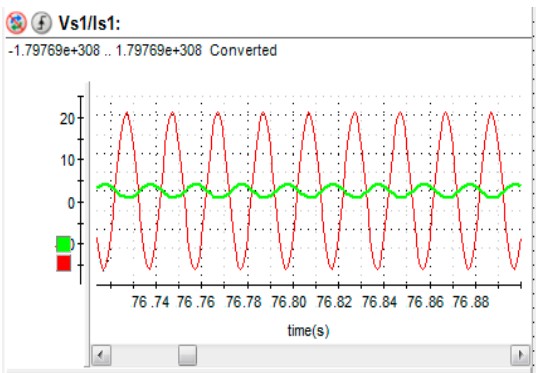

**Figure 19.** Voltage and current of a phase.

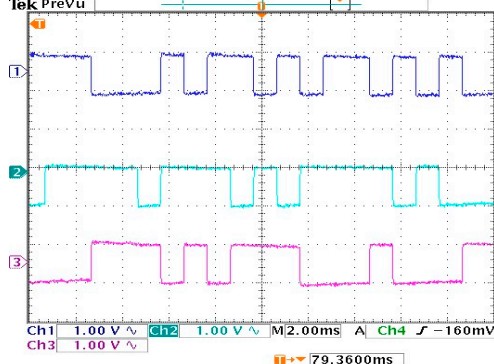

**Figure 20.** MLI signals.

The experimental and simulated data must be overlapped and compared simultaneously to conduct a thorough quantitative and qualitative analysis. The resulting experimental data were imported from the Control Desk and then charted. Figures 21 and 22 exhibit the comparison between simulated and experimental results.

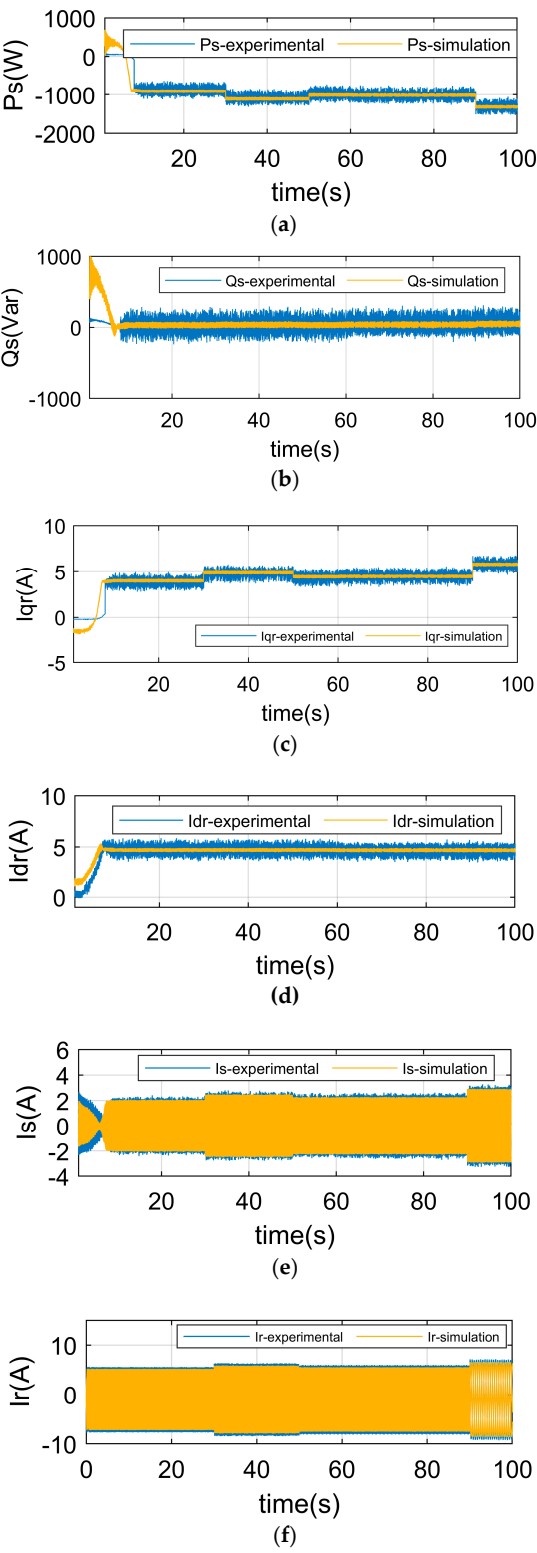

**Figure 21.** +Simulated+and+experimental+results+for+step+profile. (**a**) +active+power+; (**b**) +reactive+power+; (**c**) quadrature rotor current; (**d**) direct rotor current; (**e**) stator currents; (**f**) rotor currents.

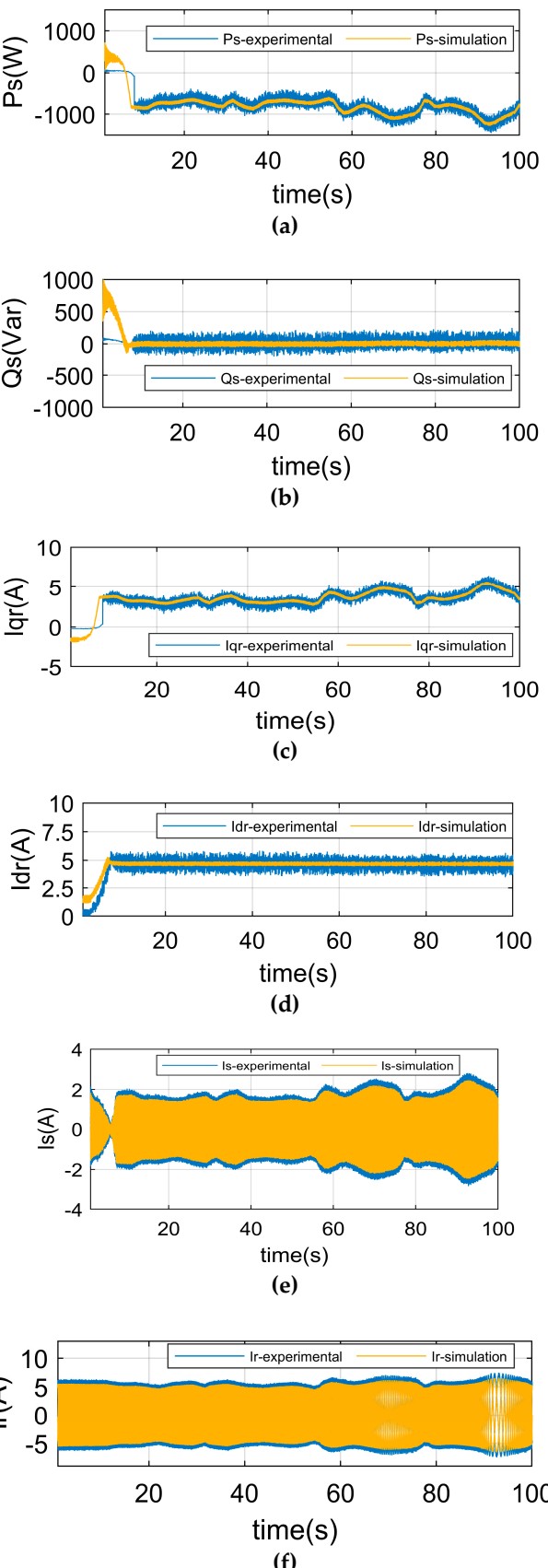

**Figure 22.** Simulated+and+experimental+results+for+wind+profile. (**a**) +active+power+; (**b**) +reactive+power+; (**c**) quadrature rotor current; (**d**) direct rotor current; (**e**) +Stator+currents+; (**f**) rotor currents.

Figures 21a,b and 22a,b exhibit active and reactive power tracking and the simulations that follow them. The error between experimental and simulation findings was calculated using each critical parameter's root mean square error for the step and the real wind profile. Ps, Qs, Iqr, and Idr are tracking their references with an inaccuracy of 8.3%, 6,1%, 8%, and 8.3%, respectively, for the step wind profile. The appropriate tracking errors for the reel wind profile are 6%, 2.9%, 2.6%, and 5.5%, respectively (Figure 23).

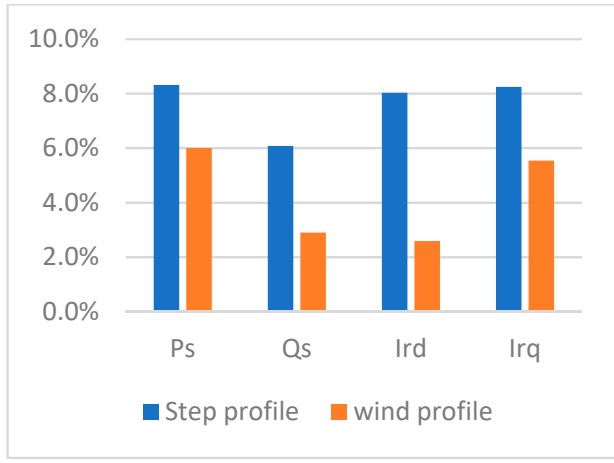

**Figure 23.** Experimental RMSE.

However, these errors are tolerable in experimental manipulation and have a negligible effect on the suggested control scheme's performance, as illustrated in Figure 23. The low error rates demonstrate the success of the experimental test unequivocally.

Table 1 represents a comparison between the RMSEs calculated from the results found through simulation and experiment for the two profiles studied. However, the difference between experimental and simulation errors are tolerable and have a negligible effect on the suggested control scheme's performance.

**Table 1.** RMSE comparison.

|  | Step Profile | | Real Wind Profile | |
| --- | --- | --- | --- | --- |
|  | **Simulation** | **Experimental** | **Simulation** | **Experimental** |
| Ps | 7.7% | 8.3% | 4.2% | 6% |
| Qs | 10.5% | 6.1% | 9.9% | 2.9% |
| Ird | 9.7% | 8% | 7.5% | 2.6% |
| Irq | 8.1% | 8.3% | 4.7% | 5.5% |

## 6. Conclusions

The originality of this study was to develop a new wind emulator and a power control strategy for a wind system based on a doubly fed induction generator. Initially, the control technique was detailed in full. Then, both converters were fitted with it (machine and grid sides). Three stages were used to evaluate the control solution: (1) a MATLAB/Simulink simulation to validate the reference's persistence (for both real and step wind speeds) and robustness, (2) real-time implementation on a dSPACE-DS1104 board connected to an experimental laboratory bench, and (3) overlapped comparison experimental and simulated data to conduct a thorough quantitative and qualitative analysis using root-mean-square error measures. The simulation and experimental results reveal that the proposed model is viable and exhibits an excellent correlation between experimental and simulated results. The future focus will be on grid-side power converter control via developing a new grid emulation design.

**Author Contributions:** Conceptualization, M.B.; methodology, M.B.; software, M.B.; validation, M.B., B.B. and H.A.A.; formal analysis, M.B.; investigation, M.B.; resources, M.B.; data curation, M.B.; writing—original draft preparation, M.B.; writing—review and editing, M.B., M.K.; visualization, M.B.; supervision, M.B., B.B., M.K.; project administration, B.B. All authors have read and agreed to the published version of the manuscript.

**Funding:** This research received no external funding.

**Institutional Review Board Statement:** Not applicable.

**Informed Consent Statement:** Not applicable.

**Acknowledgments:** This work would not have been possible without the technical support of LGEM laboratory at Mohamed First University where the experimental works were carried out.

**Conflicts of Interest:** The authors declare no conflict of interest.

## Nomenclatures

| | |
|---|---|
| DFIG | Doubly Fed Induction Generator |
| WECS | Wind Energy Conversion System |
| MPPT | Maximim Power Point Tracking |
| SCIG | Squirrel Cage Induction Generator |
| DC | Direct Current |
| AC | Alternative Current |
| FOC | Field Oriented Control |
| RSC | Rotor Side Control |
| GSC | Grid Side Control |
| DSP | Digital Signal Process |
| RTI | Real-Time Interface |
| THD | Total Harmonic Distortion |
| GUI | Graphical User Interface |
| $P_v$ | Wind Power |
| $\lambda$ | Tip Speed Ratio |
| $\beta$ | Pitch Angle |
| $C_p$ | Power Coefficient |
| LGEM | Electrical Engineering and Maintenance Maboratory |
| S | Surface Swept By The Blade |
| V | Wind Speed |
| $\omega_t$ | Turbine Speed |
| $R_s$, $R_r$, $R_f$ | Stator, Rotor, Filter Resistance |
| $L_s$, $L_r$, $L_f$, $L_M$ | Stator Rotor, Filter Mutuel Inductance |
| $V_{sd}$, $V_{sq}$ | (d-q) Axes Stator Voltages |
| $V_{rd}$, $V_{rq}$ | (d-q) Axes Rotor Voltages |
| $V_{gd}$, $V_{gq}$ | (d-q) Axes Grid Voltages |
| $I_{sd}$, $I_{sq}$ | (d-q) Axes Stator Currents |
| $I_{rd}$, $I_{rq}$ | (d-q) Axes Rotor Currents |
| $I_{gd}$, $I_{gq}$ | (d-q) Axes Grid Currents |
| $P_s$, $P_r$, $P_g$ | Stator, Rotor, Grid Active Power |
| $Q_s$, $Q_r$, $Q_g$ | Stator, Rotor, Grid Reactive Power |
| $\varphi sd$, $\varphi sq$ | (d-q) Axes Stator Fluxes |
| $V_{dc}$ | Dc Link Voltage |
| $P_{inv}$, $P_{rec}$, $P_f$ | Inverter, Rectifier, Filter Powers |
| $I_{inv}$, $I_{rec}$ | Inverter, Rectifier Currents |

## Appendix A

**Table A1.** Wind turbine parameters.

| DFIG Parameters | | WT Parameters | |
|---|---|---|---|
| Pn | 1.5 kW | R | 2 |
| p | 2 | $\rho$ | 1.22 kg/m$^3$ |
| Rs | 4.85 $\Omega$ | $\lambda$opt | 9.1 |
| Rr | 3.805 $\Omega$ | Cp | 0.5 |
| Ls | 274 mH | G | 3 |
| Lr | 258 mH | | |

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
