# Peer review of "Real-Time Power Control of Doubly Fed Induction Generator Using Dspace Hardware"

_sustainability, doi:10.3390/su15043638_

Round 1

Reviewer 1 Report

The language of the manuscript is good, however, there are a few grammatical and typo errors to be corrected.

Please add numbering to last paragraph of conclusions.

Please add to your manuscript, according to the guide for authors, the following: Funding Information, Author Contributions, Conflict of Interest and other Ethics Statements

Enhance your literature review with research papers in the field of interest.

Please explain the selection of the wind step profile from Figure 10.

Author Response

Response to Reviewer 1 Comments

The language of the manuscript is good, however, there are a few grammatical and typo errors to be corrected.

We thank the Reviewer for her/his efforts in reviewing our manuscript and for providing an overall positive evaluation of the theoretical and methodological contents.

Please add numbering to last paragraph of conclusions.

Thank you very much for this remark, thus the authors added the numbering.

Please add to your manuscript, according to the guide for authors, the following: Funding Information, Author Contributions, Conflict of Interest and other Ethics Statements

Thank you for pointing this out.  We agree with the Reviewer, The authors made the necessary change according to the guide for authors, and the details are shown as follow:

  1. Patents

Author Contributions: “Conceptualization, B.M.; methodology, B.M.; software, B.M.; validation, B.M., B.B. and H.A.; formal analysis, B.M.; investigation, B.M.; resources, B.M.; data curation, B.M.; writing—original draft preparation, B.M.; writing—review and editing, B.M.; visualization, B.M.; supervision, B.M., B.B; project administration, B.B.. All authors have read and agreed to the published version of the manuscript.”

Funding: “This research received no external funding”

Institutional Review Board Statement: OR “Not applicable”

Informed Consent Statement: “Not applicable.

Acknowledgments: This work would not have been possible without the technical support of LGEM laboratory at Mohamed First University where the experimental works have been done

Conflicts of Interest: “The authors declare no conflict of interest.”

Enhance your literature review with research papers in the field of interest.

Thank you very much for this remark , We have made extensive corrections in the literature review section according to the reviewer's suggestions. In fact, we enhanced the literature review with research papers in the field of interest. The details are shown as follow:

“…Validating and enhancing the control of wind turbines requires a test bench environment. Wind turbines are known to exhibit nonlinear behavior. A wind turbine emulator is an essential tool for modeling a real wind turbine's static, dynamic, and nonlinear properties without relying on available natural wind resources or commercial wind turbines. Authors in [19] and [20] used the Dspace card to realize a hardware under-loop simulation. As a perspective in their study, authors in [21] plan to conduct experimental tests after realizing the hardware in the loop. ….”

[19]       N. El Ouanjli et al., “Real-time implementation in dSPACE of DTC-backstepping for a doubly fed induction motor,” Eur. Phys. J. Plus, vol. 134, no. 11, 2019, doi: 10.1140/epjp/i2019-12961-x.

[20]       E. M. Youness et al., “Implementation and validation of backstepping control for PMSG wind turbine using dSPACE controller board,” Energy Reports, vol. 5, pp. 807–821, 2019, doi: 10.1016/j.egyr.2019.06.015.

[21]       S. Mensou, A. Essadki, I. Minka, T. Nasser, and B. B. Idrissi, “Control and Hardware Simulation of a Doubly Fed Induction Aero-Generator Using dSPACE Card,” Proc. 2019 Int. Conf. Comput. Sci. Renew. Energies, ICCSRE 2019, no. 1, pp. 1–7, 2019, doi: 10.1109/ICCSRE.2019.8807498.

[22]       S. Mensou, A. Essadki, I. Minka, T. Nasser, B. B. Idrissi, and L. Ben Tarla, “Performance of a vector control for dfig driven by wind turbine: Real time simulation using DS1104 controller board,” Int. J. Power Electron. Drive Syst., vol. 10, no. 2, pp. 1003–1013, 2019, doi: 10.11591/ijpeds.v10.i2.1003-1013.

Please explain the selection of the wind step profile from Figure 10.

Thank you for pointing this out.  In order to evaluate the control method, the authors used two different wind profiles. The first profile(figure 10) is a step profile to simulate abrupt changes in the machine to assess its robustness.  

Reviewer 2 Report

The novelty of this work aims to design a new wind emulator and design a power control approach for a Doubly Fed Induction Generator based wind system. Minor revisions are recommended before publication. The specific comments are as follows.

1. The DFIG in the abstract needs to be clearly expressed in terms of what it means. The suggested format is: Doubly Fed Induction Generator (DFIG). This will make it easier to read.

2. Citations need to be standardized. For example, [36]

3. Figure names need to be standardized. For example Fig.1 or Figure 1.

4. Figures 17-19 are not clearly presented, and it is recommended that the data be extracted and redrawn. And other similar figures in the paper should be corrected.

5. The coordinate titles of all figures need to be in a uniform font. The content of all images in the article needs to be modified.

Reviewer 3 Report

1.       Some typographical and grammatical errors of this paper should be carefully checked and corrected.

2.       Motivation of this work should be properly mentioned in the introduction section.

3.       It is suggested to presents some results in tabular form for better understanding and comparison purpose.

4.       Journal Standard reference format need to be followed for this paper.

Reviewer 4 Report

Thanks for the esteemed authors for their submission. The paper presents useful contribution for the field of real time control of wind turbines. However, those comments are suggested to improve the paper originality and clarity:

- Modify the title to include "Power" not "Powers".

- State clear definition for the Pitch angle in the introduction/background section before the literature review. 

- What is the value added over what has been published in reference [34]? and this reference as well https://ieeexplore.ieee.org/document/8807498

- Reword the contribution clearly according to point 3 that I mentioned. 

- What are the MATLAB solver and the step size required for real time simulation?

- In figure.8,  I think the arrow direction should be from DFIG to the inverter, not the reverse. Incorrect figure drawing should be fixed and check carefully other figures. 

- Reorganize the paper into sustainability MDPI template and follow the guidelines.

- Figures 12 and 13 should be preferably on a landscape page. 

Best wishes for the authors 

Round 2

Reviewer 4 Report

The authors have made significant effort to modify the paper. I recommend acceptance in present form.